# Trends in Tobacco Use among Children and Adolescents in Israel, 1998–2015

**DOI:** 10.3390/ijerph17041354

**Published:** 2020-02-20

**Authors:** Riki Tesler, Tanya Kolobov, Liat Korn, Kerem Shuval, Diane Levin-Zamir, Adilson Marques, Yossi Harel Fisch

**Affiliations:** 1The Department of Health Systems Management, Ariel University, Ariel 4076405, Israel; liatk@ariel.ac.il; 2School of Education, Bar Ilan University, Ramat Gan 5290002, Israel; tanyak@ariel.ac.il (T.K.); harelyossi@gmail.com (Y.H.F.); 3School of Public Health, University of Haifa, Haifa 3498838, Israel; kerem.shuval@gmail.com (K.S.); diamos@zahav.net.il (D.L.-Z.); 4Department of Health Education and Promotion, Clalit Health Services, 101 Arlozorov St. Tel Aviv 62098, Israel; 5Faculdade de Motricidade Humana, Universidade de Lisboa, Estrada da Costa, 1499-002 Cruz Quebrada, Dafundo, Portugal; amarques@fmh.ulisboa.pt

**Keywords:** cigarette smoking, grade, sex, ethnicity, adolescence, trends

## Abstract

Objectives: This study aims to measure trends in cigarette smoking among children and adolescents in Israel, focusing on school grade, sex, and ethnicity. We hypothesized that smoking would be higher among boys and Arab-Israelis, rates would grow with age, and there would be a decline over time. Methods: Data were derived from the Health Behavior in School-aged Children study between 1998 and 2015 in Israel. The total sample included 56,513 students in grades 6, 8, and 10, with 29,411 girls and 27,102 boys. Descriptive analysis described trends of smoking behavior according to grade, sex, ethnicity, family affluence, and year of study. multivariate logistic regression analysis examined predicting variables. Results: Smoking was higher among boys in all grades, ethnic groups, and years of study, with the highest frequencies among Arab-Israelis. Trends over the years show a decline from 1998 to 2004, followed by an increase for both sexes. The increase was more prominent among girls. Logistic regression analysis revealed strong associations between smoking and grade, sex, ethnicity, and year of study. Conclusions: The results of this study can significantly enhance the development and implementation of smoking prevention and control programs among students in Israel.

## 1. Introduction

Tobacco use is the world’s leading cause of preventable morbidity and mortality. Resulting in nearly 6 million deaths each year, tobacco use imposes an onerous burden on society [1]. The tobacco epidemic continues, despite decades of research and extensive knowledge about the consequences [2]. Cigarette smoking is a dangerous behavioral health pattern, often causing serious illnesses responsible for morbidity and mortality,^2^ including cancer, cardiovascular diseases, and respiratory problems [3,4]. Smoking among adolescents has been associated with eating disorders, alcohol consumption, bullying, drug consumption, and delinquency [5,6,7]. Engaging in such risky behavior is often a means to gain recognition, control, and a sense of independence; however, it could lead to habitual behavior and adverse health outcomes [6,7,8].

In Israel, there is a disparity in smoking rates between the ethnic minority (Arab-Israeli) to the ethnic majority (Jewish-Israeli) population [9,10,11]. According to the Israeli Ministry of Health, the rate of smoking in 2014 among Arab-Israelis is higher than that of Jewish-Israelis: 26.3% vs. 18.4% [12]. Thus, there is a need to focus tobacco control efforts not only on the entire Israeli population, but among the ethnic minority in particular [11]. In the past decade, there has been an effort to implement such tobacco control programs in the school setting in Israel [9]. The percentage of 6th, 8th, and 10th graders who smoke at least once a week is 3.5% among Jewish-Israelis compared to 4.14% among Arab-Israelis; boys reported higher rates than girls, and about 20% of 10th graders reported smoking. This rate is slightly higher than the average for this age group in Europe, which is approximately 17% [13].

In the current study, we examined whether smoking trends in Israel changed from 1998 to 2015 with a focus on grade, sex and ethnicity. We hypothesized that boys would have higher smoking rates than girls, Arab-Israelis would have higher rates than Jewish-Israelis, smoking rates would increase with age, and there would be a decline during the years. 

## 2. Methods

Data for this study were derived from the Health Behavior in School-aged Children (HBSC) survey, a World Health Organization collaborative cross-national study conducted in 47 countries, including Israel. The HBSC study aims to describe adolescent health, behaviors, and psychosocial determinants for health behaviors. It consists of surveys based on a standardized international protocol, carried out among country-representative samples of students from 6th, 8th and 10th grades [14].

The Israeli samples of the HBSC were all nationally representative. Data regarding smoking behavior over a 16 year time frame (1998–2015) were collected in six different years. For the current study, we utilize all data on Israeli students collected in Israel. Sample size in each survey year varies by sex and ethnicity (Table 1).

The sampling procedure is described in detail elsewhere [13,14]. Briefly, all general public schools in Israel were eligible for inclusion in the study. The majority of students in Israel study in public schools; there are few private schools throughout the country. Selection of the schools in the strata is random. All students in selected classes attending school on the day of the survey were eligible to participate. Data were collected in anonymous self-report questionnaires which were made available in Hebrew and Arabic. The total sample included 56,513 participants who participated in all six samples between 1998 and 2015. The Israeli HBSC research protocol received approval from the research ethics committees of the Israeli Ministry of Education, Bar Ilan University, and Ariel University.

### 2.1. Variables 

Sociodemographic variables were derived from self-reported responses to questions regarding grade (6th, 8th, or 10th grade), sex, ethnicity (Jewish-Israeli or Arab-Israeli), and family affluence (based on score that is described). A family affluence scale (FAS) was created based on the following six items: 1) ‘Does your family own a car, van, or truck?’ (‘No’ = 0, ‘Yes, one’ = 1 and ‘Yes, two or more’ = 2); 2) ‘Do you have your own bedroom?’ (‘No’ = 0 and ‘Yes’ = 1); 3) ‘During the past 12 months, how many times did you go on holiday with your family?’ (‘Not at all’ = 0, ‘Once’ = 1, ‘Twice’ = 2, and ‘More than twice’ = 3); 4) ‘How many computers does your family own?’ (‘None’ = 0, ‘One’ = 1, ‘Two’ = 2, and ‘More than two’ = 3); 5) ‘How many baths/showers are there in your house?’ (‘None’ = 1, ‘One’ = 2, ‘Two’ = 3 and ‘More than two’ = 4); and 6) ‘Does your family have a dishwasher at home?’ (‘No’=1, ‘Yes’ = 2). The reliability test showed Alpha Cronbach to be α = 0.72 for these six items. A new scale variable was calculated by summing up the scores of all six items. For the purpose of the current study, FAS was categorized into three categories: 0–6 indicated a low level of family affluence, 7–9 indicated a medium level of family affluence, and 10–13 indicated a high level of family affluence; this was same scale as reported in Currie et al., 2008 [14].

To create a lifetime smoking variable, the following question was posed: ‘How many days (if any) have you ever smoked cigarettes?’ (‘Never’, ‘1–2 days’, ‘3–5 days’, ‘6–9 days’, ‘10–19 days’, ‘20–29 days’, or ’30 days or more’.) A dichotomous variable was created, where 1 or more = ‘Ever tried to smoke’ and 0 = ‘Never smoked’. The everyday smoking variable was defined based on the following question: ‘How often do you currently smoke tobacco?’ (‘Every day’, ‘At least once a week, but not every day’, ‘Less than once a week’, or ‘Never’.) Adolescents who reported smoking cigarettes ‘every day’ were considered daily cigarette smokers, while the other categories were grouped together and regarded as non-daily smokers [15]. 

### 2.2. Statistical Analysis

Descriptive statistics were used to examine trends in smoking among youth by grade and sex, stratified by ethnicity. Moreover, multivariate logistic regression was employed in separate models for the two ethnicities to predict smoking, stratified by grade, sex, and FAS. 

### 2.3. Human Subjects Approval Statement

The Israeli HBSC research protocol received approval from the research ethics committees of the Israeli Ministry of Education, Bar Ilan University, and Ariel University.

## 3. Results

Israeli adolescents’ smoking trends from 1998 to 2015 are presented in Figure 1 and Figure 2, and multivariate regression results are depicted in Table 1 and Table 2. Figure 1a–c show the prevalence of ever smokers among adolescents in the 6th, 8th, and 10th grades by sex and ethnicity, according to the survey year. Accordingly Figure 2a–c show the prevalence of daily smokers among adolescents by grade, sex, and ethnicity according to survey year.

As seen in Figure 1, in both ethnicities, the prevalence of ever smokers was higher among boys than girls across all grades and years of study. Compared with 1998, in 2015, in each grade, ever smoking rates decreased among Jewish boys and girls, whereas among Arab youth, an increase was observed. For example, among Jewish boys, the ever smoking rate in 6th grade was 7.1% in 1998 and decreased to 4.7% in 2015; in 8th grade it decreased from 15.7% in 1998 to 6.8% in 2015; and in 10th grade it decreased from 31.6% in 1998 to 18.2% in 2015. Among Jewish girls, the rate of smoking in 6th grade decreased from 2.6% in 1998 to 1.9% in 2015; from 13.8% in 1998 to 2.9% in 2015 in 8th grade; and from 21.8% in 1998 to 8.8% in 2015 in 10th grade. Among Arab students of both genders, there was a similar and constant increase, with 10th grade Arab boys with rates of 41.5% in 2011 and 41.4% in 2015. Ever smoking rates among Arab boys were almost always higher than among all other groups in all grades (with the exception of a similar rate to 10th grade Jewish boys in 1998 and 2004). In 2015, the ever smoking prevalence among Jewish boys in 6th grade was 4.7%, while it was 26.3% among Arab boys. In the same year, the difference was quite similar among 8th grade students, at 6.8% for Jewish boys and 26.2% for Arab boys. In the same year, the smoking prevalence in the 10th grade was higher for both ethnicities, but the difference became even wider at 18.2% for Jewish boys and 41.4% for Arab boys. The general trend of ever smoking among Arab boys also rose between 1998 and 2015. In 6th grade, it rose from 19.6% to 26.3%; in 8th grade, it rose from 22.9% to 26.2%; in 10th grade, it rose from 31.5% to 41.4%.

Another trend was an increase in the ever smoking prevalence of Arab girls as can be seen in Figure 1b,c, especially in 8th and 10th grades. In 2011, the rate of ever smoking among Arab girls in 8th grade was 9.2%, in comparison to 3.1% in 2006 and 2.6% in 2004. In the 10th grade the prevalence of ever smoking among Arab girls was 10.5% in 2011, compared to 4.2% in 2006 and 4.5% in 2004. The trend of smoking among Jewish girls in these same years saw a decline, from 4.5% in 2006 to 2.6% in 2011 in 8th grade, and from 13.9% in 2006 to 8.3% in 2011 in 10th grade.

Figure 2a–c present prevalence of daily smoking in each grade by sex, ethnicity, and survey year. Within each ethnicity across all grades, daily smoking rates were higher among boys compared to girls. In the 6th and 8th grades, the rate of daily smokers was higher among Arab boys compared to all groups in all survey years. In 10th grade this prevalence was higher among Jewish boys until 2006, when Arab boys again had the highest rates. Among the 10th grade boys, the prevalence of daily smoking in 2004 was 14.3% among Jewish boys and 11.3% among Arab boys; in 2011 it became 10.0% among Jewish boys and 19.6% among Arab boys.

The general tendency of daily smoking in the 6th grade was relatively stable for Jewish boys, Jewish girls, and Arab girls, but among Arab boys. The findings indicated an increase from 6.4% in 1998 to 9.8% in 2015. Jewish boys and girls in the 8th grade saw a declining trend in daily smoking from 1998 (boys—5.8%; girls—2.7%) to 2015 (boys—3.7%; girls—1.3%). For Arab boys in the 8th grade, the tendency to smoke daily was more complex: the rate decreased from 9.0% in 1998 to 4.5% in 2006 and then rose to 7.0% in both 2011 and 2015. A similar pattern was found among Arab girls, but the rate of daily smoking fell from 1.9% in 1998 to 0.4% in 2004 and then rose from 1.2% in 2006 to 3.9% in 2015.

Daily smoking trends for Jewish 10th graders declined from 1998 (boys—17.2%; girls—7.9%) to 2015 (boys—9.0%; girls—3.4%). The general tendency for Arab boys and girls also witnessed a decline in daily smoking rates from 1998 (boys—12.3%; girls—3.3%) to 2006 (boys—10.5%; girls—0.3%) and then an increase to 19.6% among boys and 5.6% among girls in 2011, and then to 19.4% among boys and 5.7% among girls in 2015.

Table 2 presents the results of the logistic regression that was performed to predict lifetime smoking among adolescents according to grade, sex, ethnicity, and FAS by survey year. Differences between boys and girls expanded over the years among Jewish adolescents. For example, in 1998 the adjusted odds ratio (AOR) of boys who smoked daily compared to girls was 1.61 (95% confidence interval (CI): 1.31–1.99), but in 2015 it rose to 2.92 (95% CI: 2.29–3.72). For Arab adolescents, the ORs were higher than Jewish students; in 2015, the AOR for daily smoking among Arabs was 4.23 (95% CI: 3.25–5.49) while it was 2.92 (95% CI: 2.29–3.72) among Jews.

The pattern of gender-based disparities among Arab adolescents was different than the trend among Jewish students. The odds were higher in 2004 and 2006 (approximately AOR: 7) in comparison to other years (approximately AOR: 4). The pattern of the influence of grade over the years also differed between Jewish and Arab adolescents; among Jews, the odds ratio (OR) decreased from 2.58 (95% CI: 2.26–2.95) in 1998 to 1.92 (95% CI: 1.67–2.22) in 2015. In contrast, among Arabs, the AOR stayed stable (approximately AOR: 1.3). In both groups, FAS had almost no influence on predicting lifetime smoking (about AOR: 1).

The sociodemographic predictors of daily smoking among adolescents are presented in Table 3. Among Jewish students, the odds of daily smokers to be male were higher in 2011 (AOR: 3.11, 95% CI: 2.26–4.28, p < 0.001) and 2015 (AOR: 3.04, 95% CI: 2.28–4.07) in comparison to earlier years (e.g., AOR: 1.46, 95% CI: 1.10–1.94, p < 0.001, in 2002) among Jewish students. For Arab adolescents, the OR for boys to be daily smokers was generally around 3 and was especially high in 2004 (AOR: 11.85 95% CI: 7.60–18.49, p < 0.001). The impact of grade on daily smoking among Jewish students increased from OR: 2.63 (95% CI: 2.15–3.22) in 1998 to AOR: 4.51 (95%CI: 3.58–5.67) in 2004 and then decreased to AOR: 1.94 (95% CI: 1.64–2.30) in 2015. Grade-based differences in daily smoking were lower among Arab students (AOR: 1.29, 95% CI: 1.03–1.61 in 1998; AOR: 1.44, 95% CI: 1.19–1.74 in 2015), and did not vary much over the years (Table 3). Family affluence among both groups had almost no influence on daily smoking (approximately AOR: 1).

## 4. Discussion

Scant evidence exists from large national samples of Israelis pertaining to cigarette smoking levels among adolescents in Israel; the pool of adolescents created by the HBSC survey was crucial to our study. Our findings from this large representative national sample of adolescents in Israel indicate several essential results: boys were more likely to smoke than girls, and there were differences in levels of smoking between Arab-Israelis and Jewish-Israelis, with the former primarily engaging in more smoking. Hence, there was a disparity favoring the ethnic minority in terms of higher smoking levels. 

Results of this study indicate that there were marked reductions in smoking between 6th, 8th, and 10th grades. This is consistent with prior studies that found that substance abuse declines and smoking increases between ages 10 and 12 years and later teenage years [3,4,5]. High levels of cigarette smoking during adolescence can contribute to poor health outcomes in adulthood like depression, and psychosomatic symptoms [6,7,8]. In a systematic review that studied predictors of smoking onset, Wellman et al. also found that with increased age and grade, there was an association with an increased risk of smoking onset [16]. We found a decline in everyday smoking in 2006, which could possibly be due to the fact that in 2005, the Framework Convention on Tobacco Control was entered into force globally. This framework aims to reduce tobacco use, by implementing tax, advertising, and promotional restrictions on tobacco products [17].

Factors related to adolescents’ sex were also found to be potential determinants on smoking. We found that boys were more likely than girls to smoke in all grades, in the two ethnic groups, and years of study. National surveys from the United States have indicated that smoking could vary by sex, with adolescent males initiating and smoking cigarettes at higher rates than females [18]. Additional studies have confirmed that males are more likely than females to smoke cigarettes [13,18,19,20,21]. Romer (2012) pointed out that there could be various contributors to differences in substance use patterns between girls and boys, including biological factors and socialization processes [22].

As previously stated, our study found differences in smoking trends between the two ethnicities. Cigarette use was higher among the ethnic minority, Arab-Israelis, than among Jewish-Israelis. This is consistent with other research. National data from Israel indicate that the Arab population is 1.6 times more likely to smoke cigarettes than the Jewish population [11]. In an additional study by Thompson et al. (2017), which studied time trends in smoking onset, authors found that between 2006 and 2013, there were declines among White participants as compared to Blacks and Hispanics; smoking onset rates increased among Hispanic and Black participants [23].

The limitations of the HBSC study include a repeated cross-sectional design. This limits causal inferences. The HBSC sampling strategy excludes adolescents in non-classroom settings, which may also impact the external validity of our findings. Additionally, our findings only pertain to students from public schools in Israel, as the Ministry of Health approved this study be undertaken in public and not private schools. In Israel, the vast majority of students study in public schools; there are not many private schools, so we believe our results were not impacted by focusing only on public schools. Furthermore, our study relies on answers obtained from self-reported questionnaires filled out by students, and therefore, responses may be biased, as some students may not have answered truthfully or could have underreported the number of cigarettes smoked. To try to limit untruthful answers, students were assured prior to filling out questionnaires that neither their teachers nor parents would view answers, and that they would remain anonymous. To strengthen this doubt, previous research has shown that self-reported answers regarding the prevalence of smoking is a good indicator of the actual smoking status [15]. Finally, as this study was conducted in Israel, the implications of the findings may not be relevant in other settings.

In summary, to our knowledge, there has not been a previous analysis of Israeli adolescents stratified by sex and ethnicity, as well as key contextual factors. This study’s results illuminate disparities and the importance of promoting smoking prevention among adolescents. This research allowed us to systematically monitor the use of tobacco among students, which is the first step in the planning of prevention strategies. The aim of reducing health inequities needs to be a key part of a comprehensive strategy when discussing health promotion and development [24]. When designing effective tobacco prevention and control programs, one must enumerate individual smoking determinants such as grade, sex, ethnicity, and FAS. Our calibrated results engender important policy implications for the development of cigarette smoking prevention programs for Israeli youth, specifically in the school environment.

## Figures and Tables

**Figure 1 ijerph-17-01354-f001:**
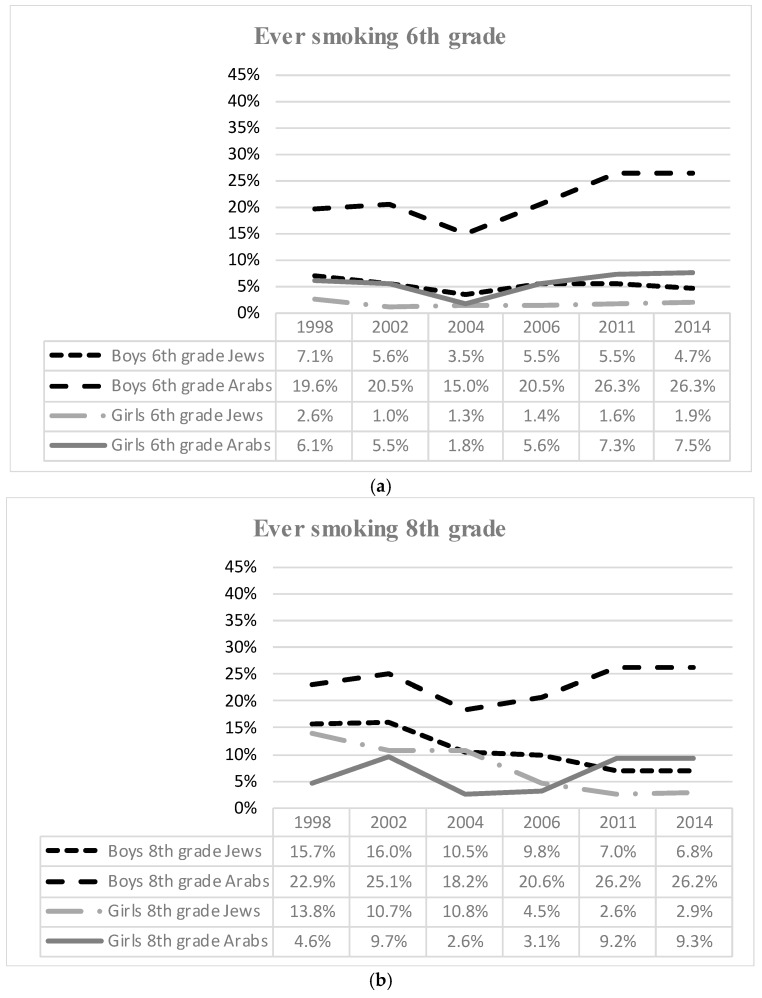
(**a**) Ever smoking 6th grade; (**b**) Ever smoking 8th grade; (**c**) Ever smoking 10th grade.

**Figure 2 ijerph-17-01354-f002:**
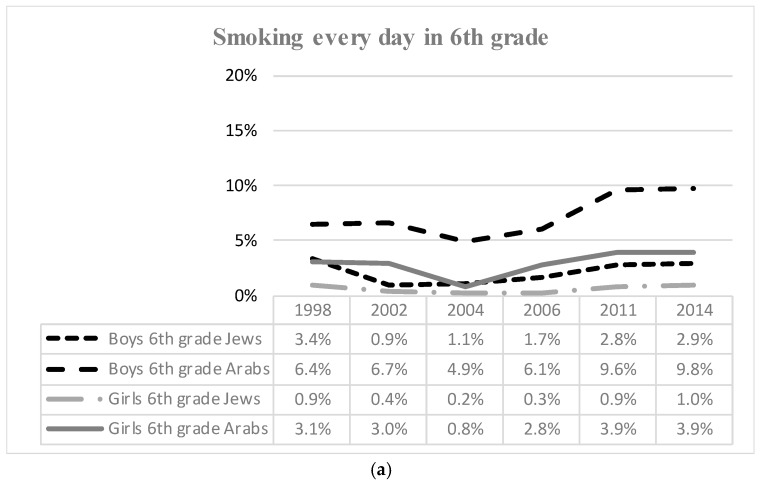
(**a**) Daily smoking 6th grade; (**b**) Daily smoking 8th grade; (**c**) Daily smoking 10th grade.

**Table 1 ijerph-17-01354-t001:** Sample size by survey year, ethnicity and gender subgroups, Israel, 1998–2015.

Year	Jewish Students			Arab Students			Total Sample		
	Boys	Girls	Total	Boys	Girls	Total	Boys	Girls	Total
1998	2685	2603	5288	1505	1601	3106	4190	4204	8394
2002	2252	2539	4791	631	737	1368	2883	3276	6159
2004	2447	2808	5255	2779	3254	6033	5226	6062	11,288
2006	1792	2293	4085	1105	1423	2528	2897	3716	6613
2011	4205	4130	8335	1501	1717	3218	5706	5847	11,553
2015	4699	4589	9288	1501	1717	3218	6200	6306	12,506
**Total**	**18,080**	**18,962**	**37,042**	**9022**	**10,449**	**19,471**	**27,102**	**29,411**	**56,513**

**Table 2 ijerph-17-01354-t002:** Results from logistic regression to predict lifetime smoking, adjusted odds ratios by gender, grade, and family affluence, presented by ethnicity and studied period (1998–2015).

Year	Ethnicity							
	Jews				Arabs			
	Gender (Boys = 1)	Grade	Family Affluence	R^2^	Gender (Boys = 1)	Grade	Family Affluence	R^2^
	AOR (95% CI)	AOR (95% CI)	AOR (95% CI)		AOR (95% CI)	AOR (95% CI)	AOR (95% CI)	
1998	1.61 * (1.31–1.99)	2.58 * (2.26–2.95)	1.00 (0.93–1.07)	0.13	4.75 * (3.57–6.32)	1.28 * (1.10–1.49)	0.98 (0.90–1.06)	0.12
2002	1.57 * (1.32–1.88)	2.54 * (2.23–2.88)	0.95 (0.91–1.00)	0.11	3.68 * (2.64–5.12)	1.30 * (1.07–1.57)	1.07 (0.99–1.16)	0.09
2004	1.36 * (1.14–1.63)	3.11 * (2.73–3.54)	0.95 (0.90–1.00)	0.14	7.67 * (6.14–9.59)	1.39 * (1.25–1.55)	1.01 (0.96–1.05)	0.15
2006	1.90 * (1.51–2.40)	2.56 ^*^ (2.17–3.03)	0.94 ** (0.89–0.96)	0.10	6.65 * (4.88–9.04)	1.09 (0.93–1.28)	1.02 (0.96–1.08)	0.15
2011	3.01 * (2.41–3.76)	2.32 * (2.03–2.66)	0.92 * (0.89–0.96)	0.11	4.73 * (3.76–5.93)	1.33 * (1.17–1.52)	0.96 (0.92–1.00)	0.13
2015	2.92 * (2.29–3.72)	1.92 * (1.67–2.22)	0.92 * (0.88–0.96)	0.08	4.23 * (3.25–5.49)	1.38 * (1.18–1.60)	0.95 ** (0.90–0.99)	0.11

Note: Abbreviations: R^2^, logistic regression; AOR, adjusted odds ratio; CI, confidence interval; * p < 0.05; ** p < 0.01.

**Table 3 ijerph-17-01354-t003:** Results from logistic regression to predict daily smoking, adjusted odds ratios by gender, grade, and family affluence, presented by ethnicity and studied period (1998–2015).

Year	Ethnicity							
	Jews				Arabs			
	Gender (Boys = 1)	Grade	Family Affluence	R^2^	Gender (Boys = 1)	Grade	Family Affluence	R^2^
	AOR (95% CI)	AOR (95% CI)	AOR (95% CI)		AOR (95% CI)	AOR (95% CI)	AOR (95% CI)	
1998	2.57 * (1.85–3.57)	2.63 * (2.15–3.22)	0.96 (0.86–1.06)	0.12	3.15 * (2.09–4.77)	1.29 * (1.03–1.61)	1.03 (0.91–1.16)	0.05
2002	1.46 * (1.10–1.94)	3.9 * (3.11–4.95)	0.95 (0.88–1.03)	0.13	2.96 * (1.75–5.03)	1.23 (0.91–1.66)	1.08 (0.95–1.23)	0.04
2004	1.55 * (1.20–2.01)	4.51 * (3.58–5.67)	0.90 * (0.84–0.97)	0.16	11.85* (7.60–18.49)	1.55 * (1.30–1.83)	1.02 (0.95–1.10)	0.14
2006	1.75 * (1.22–2.52)	3.20 * (2.38–4.31)	0.84 * (0.77–0.91)	0.11	5.77 * (3.32–10.04)	1.10 (0.84–1.44)	1.16 * (1.05–1.28)	0.10
2011	3.11 * (2.26–4.28)	2.07 * (1.72–2.50)	0.89 * (0.84–0.94)	0.09	3.03 * (2.20–4.17)	1.44 * (1.19–1.74)	0.95 (0.90–1.01)	0.06
2015	3.04 * (2.28–4.07)	1.94 * (1.64–2.30)	0.91 * (0.87–0.96)	0.08	3.03 * (2.20–4.17)	1.44 * (1.19–1.74)	0.95 (0.90–1.01)	0.06

Note: Abbreviations: R^2^, logistic regression; AOR, adjusted odds ratio; CI, confidence interval; * p < 0.01.

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
