# Peer review of "Trends in Tobacco Use among Children and Adolescents in Israel, 1998–2015"

_ijerph, 2020, doi:10.3390/ijerph17041354_

Round 1
Reviewer 1 Report
Information for Authors
Line 25-26. In the Abstract it was stated there was a strong association between family affluence and smoking; however, in the Results section (Lines 17O-171) it was stated that FAS had almost no influence on predicting lifetime smoking (about OR:1)”. Similarly in Lines 181-182, it is reported that “family affluence among both groups had almost no influence on daily smoking (approximately OR:1)”. Please review and clarify abstract. The remainder of the abstract is clear and consistent with the manuscript.
Lines 32-48. The Introduction section, although concise, provides a sufficient context in Israel compared to Europe and a global perspective among adolescents.
Lines 49-52. The sentence structure is very long. Suggest a small revision to make into at least two shorter sentences.
Line 57. Please add an “a” after on to specify the one protocol that was mentioned later.
Line 64-65. In other countries students attend both public and private schools. Please consider adding a brief comment about public and private schools in Israel to assist readers to better understand sample representativeness.
Lines 73-94. This section clearly describes the variables under consideration, their level of measurement, and satisfactory reliability of the FAS scale.
Line 97. Please clarify if the term “multivariable” logistic regression is the same as “multinomial” logistic regression. Using logistic regression given the number of variables and the levels of measurement at nominal and interval levels was an appropriate choice.
Lines 103-153. The trend data in Figures 1 and 2 is clearly described and consistently displayed. In Figure 2(a, b, c) from 2006 and onward there is a striking increase in smoking among Arab-Israeli boys and girls, particularly among smoking everyday in 10th grade.
Line 163, Table 2. Under gender (boys=1) and Year 2002 there is an “a” after 1.57. Should this be an asterisk?
Lines 212-217. The information you presented here was confusing to me and did not fit with the focus on Israeli adolescents. Please consider a revision to make this section clearer.
From the data and R2 reported in Table 2 and Table 3, the predictor variables accounted for low percentages of “lifetime smoking” and “daily smoking”. However, there is no discussion of other possible factors that might be influencing the odds of smoking among Israeli students in your study. If each of the predictor variable accounts for 9% to 16% of the variance, what other contextual factors might be at play? Particularly for the dramatic increase among Arab-Israeli students in daily smoking in Figure 2(c)? Are there cultural, religious or environmental influences in Israel that could be briefly discussed?
Lines 218-228. I agree with your stated limitations. You could consider making a statement about excluding private schools mentioned close to your statement about excluding “adolescents in non-classroom settings (Line 219).
Please double check References again for Journal formatting requirements in ACS style.
Author Response
reviewer 1
We wish to thank the reviewer for their important comments, which have substantially improved our manuscript. We have replied to each of the reviewer' points (see below). We have highlighted the corrected text in the manuscript in red print.
Information for Authors
Line 25-26. In the Abstract it was stated there was a strong association between family affluence and smoking; however, in the Results section (Lines 17O-171) it was stated that FAS had almost no influence on predicting lifetime smoking (about OR:1)”. Similarly, in Lines 181-182, it is reported that “family affluence among both groups had almost no influence on daily smoking (approximately OR:1)”. Please review and clarify abstract. The remainder of the abstract is clear and consistent with the manuscript.
Thank you for pointing this out. We deleted “family affluence” from line 26 in the abstract.
Lines 32-48. The Introduction section, although concise, provides a sufficient context in Israel compared to Europe and a global perspective among adolescents.
Thank you
Lines 49-52. The sentence structure is very long. Suggest a small revision to make into at least two shorter sentences.
You are correct, this was a long sentence. It is now two sentences.
Line 57. Please add an “a” after on to specify the one protocol that was mentioned later.
We’ve added the “a”.
Line 64-65. In other countries students attend both public and private schools. Please consider adding a brief comment about public and private schools in Israel to assist readers to better understand sample representativeness.
We have added the following sentence to this section: “The majority of students in Israel study in public schools; there are few private schools throughout the country”.
Lines 73-94. This section clearly describes the variables under consideration, their level of measurement, and satisfactory reliability of the FAS scale.
Thank you
Line 97. Please clarify if the term “multivariable” logistic regression is the same as “multinomial” logistic regression. Using logistic regression given the number of variables and the levels of measurement at nominal and interval levels was an appropriate choice.
Yes, by “multivariable”, we meant “multinominal” and changed it wherever relevant in the text.
Lines 103-153. The trend data in Figures 1 and 2 is clearly described and consistently displayed. In Figure 2(a, b, c) from 2006 and onward there is a striking increase in smoking among Arab-Israeli boys and girls, particularly among smoking every day in 10th grade.
Thank you.
Line 163, Table 2. Under gender (boys=1) and Year 2002 there is an “a” after 1.57. Should this be an asterisk?
Yes, this is supposed to be an asterisk. It has been changed. Thank you for noticing this.
Lines 212-217. The information you presented here was confusing to me and did not fit with the focus on Israeli adolescents. Please consider a revision to make this section clearer.
Thank you, after reviewing this section we deleted the sentence that did not fit and also removed it from the references list.
From the data and R2 reported in Table 2 and Table 3, the predictor variables accounted for low percentages of “lifetime smoking” and “daily smoking”. However, there is no discussion of other possible factors that might be influencing the odds of smoking among Israeli students in your study. If each of the predictor variable accounts for 9% to 16% of the variance, what other contextual factors might be at play? Particularly for the dramatic increase among Arab-Israeli students in daily smoking in Figure 2(c)? Are there cultural, religious or environmental influences in Israel that could be briefly discussed?
Thank you, we believe that we included all the relevant predictor variables.
Lines 218-228. I agree with your stated limitations. You could consider making a statement about excluding private schools mentioned close to your statement about excluding “adolescents in non-classroom settings (Line 219).
Thank you for this suggestion. We’ve added the following sentences to the limitations section: “Additionally, our findings only pertain to students from public schools in Israel, as the Ministry of Health approved this study be undertaken in public and not private schools. In Israel, the vast majority of student's study in public schools; there are not many private schools, so we believe our results were not impacted by focusing only on public schools”.
Please double check References again for Journal formatting requirements in ACS style.
Thank you. The formatting for the references has been updated to the ACS style.

Reviewer 2 Report
You can read my comments and suggestions at the attached document.

Author Response
We wish to thank the reviewer for their important comments, which have substantially improved our manuscript. We have replied to each of the reviewer' points (see below). We have highlighted the corrected text in the manuscript in red print.
Introduction:
Any study that focuses on ethnic minorities should describe the socio-economic situation of that ethnic minority, in this case the Arabs in Israel. A socio-economic situation, like that of the rest of ethnic majorities, suggests the possibility that the differences may be diluted. An example of the importance of this description can be seen in the article by Usera-Clavero et al. (Smoking Prevalence Inequalities Among Roma and Non-Roma Population in Spain Between 2006 and 2014). Please describe the socio-economic situation of the Arab-Israel minority. You declare that the samples are all nationally representative. If this is true, can you explain why the percentage of Jewish students is between 62% and 77% but in 2004 it is only 44% according to their figures? It is only representative for each ethnic group and not for the whole population of students, isn’t it?
Thank you for this important comment. As suggested by Usera-Clavero, as well as by your comment, it is true to say that adjusting the sample for the socio economic situation would probably lower the differences between Arabs and Jews. We decided to add this variable to the logistic regression, so that it would be possible to see if the socio-economic variable had an effect on these differences.
The samples are representable; we weighted them according to year for Jews and Arabs in the Israeli population for the same year.
Figures.
The figures are of poor quality. In addition, you should not mix graphs and tables. We represent the data in a graph to describe the temporal trends, but we construct a table to declare the data faithfully, in this case the percentages. I suggest deleting the data from the figures and representing the information of the percentages in a table similar to table 1. Columns: Ethnicity, Grade and Sex, rows Year: (6 rows × 12 columns).
Thank you for your suggestions, after consulting with all authors, we have decided to leave the outcomes presentations as it is.
Variables:
Regarding Family affluence, the use of Cronbach's alpha is totally not suitable, and it is an outdated indicator. It is enough to mention the authors of Family Affluence Scale (FAS). In addition, the results of this variable by Ethnicity would let us understand the context of the Arab-Israel ethnic minority.
As suggested by the reviewer we deleted the description of Cronbach's alpha.
Methodology
Regarding logistic regression, the term “stratification” should be avoided and changed to the term "adjusted by". If you have really stratified you do not need logistic regression, just calculate the OR. I understand that Tables 2 and 3 show the ORs adjusted by Gender, Grade and FAS stratified by Ethnicity and Year of study. Therefore, the title of Table 2 and 3 should change to something like: “Table 3: Results from logistic regression to predict daily smoking, adjusted odds ratios by gender, grade, and family affluence, presented by ethnicity and studied period (1998–2015)” It is a pity that the logistic regression model used has completely eliminated the possibility of having a statistical test that evaluates the differences between ethnic groups. There is no single statistical test in the manuscript that compares both ethnicities. As you describe it, the model has been adjusted for ethnicity and every single year studied. Your equation was:
ln(??)=∝0+∝1??????+∝2?????+∝3??
This model is extremely limited to describe reality.
A much more interesting model would be one that includes, at least, the Ethnicity ln(??)=∝0+∝1??????+∝2?????+∝3??+∝4
??ℎ?????? 2 And better if it also includes the interaction of ethnicity with gender. ln(??)=∝0+∝1??????+∝2?????+∝3??+∝4??ℎ??????+∝5 ??????∗??ℎ??????
We have removed the term “stratification” as suggested by the reviewer. We corrected it to the term “adjusted”, changed the titles of the tables and replaced OR to be AOR.
Since we discussed sub-populations, we decided to look at them separately.
Trend analysis.
This work does not include a trend analysis, it is simply a description. The possibilities of logistic regression to analyze the temporal trend are also highly appreciated by the scientific community. The authors can observe in the study of Usera-Clavero et al. 2019 how to use logistic regression to analyze changes in the trend between two periods in tobacco consumption in two different ethnicities. In the case of “Trends in Tobacco Use among Children and Adolescents in Israel, 1998–2015” it has 6 periods between 1998 and 2015. Therefore, there are two possibilities:
Analyze the changes year by year through logistic regression. Here we should only calculate the term of the interaction between ethnicity and year, where the year variable has become 5 dummy variables, 1 for each year and the year 1998 is directly defined. Remember that in each model you should only include the data corresponding to the two years that you intend to compare, for example 2006 vs. 2011 Analyze the changes using the Joinpoint regression. This type of analysis allows us to identify when there has been a change in trend through the percentage of annual change (PAC). It is a different approach to logistic regression but very effective. Perhaps the limitation to this regression is that your database has only 6 years, when it is recommended to have 8, although there are many authors who have applied it with fewer years of study.Thank you for very detailed suggestion. We would love to use trend analysis in a future article on this subject.
Results
Line 104. There is an error: the results of the logistic regression are shown in tables 2 and 3 not in tables 1 and 2. In the results section, avoid reading the tables again. The whole wording of the results section should be improved. My advice: try to explain the results and summarize what happens without assessing the results. Surely the reading of the tables is tedious because the models used are not the most appropriate.
Thank you for pointing this out. The mistake has been corrected, and as you advised, we summarized this section.
Discussion
The discussion is not contextualized, it should relate it to the situation of the ethnic minorities of Israel rather than compare to other realities of completely different ethnic minorities such as the US. I do not observe any comment on the fact that the ORs associated with the Grade in Jewish students are greater than those of the Arab students.
We have changed the discussion to better reflect the objective of the paper and have taken out a section that compared our results to the results of another study from the US. Additionally, we have also added that grade was more significant among Jewish students than Arabs.
The discussion of the results is really poor. You can improve your discussion by drawing on this WHO document: Whitehead M, Dahlgren G. Concepts and principles for tackling social inequities in health: levelling up Part 1. World Health Organization. 2007. pp. 2–5. ttp: //www.euro.who.int/__data/asset s / pdf_file / 0010/74737 / E8938 3.pdf.
Thank you, we have added information from this document.
Please see the attachment

Reviewer 3 Report
This paper examines trends in smoking among school -aged children in grades 6, 7 and 10 in Israel based on ethnicity and gender. Findings of this study can contribute to understanding characteristics of children who are likely to smoke cigarettes and tailor management efforts.
Authors make a good argument as to the need for this type of study and nicely show the trend in smoking rates by age and gender across a few years. This study highlights the differences among Arab-Israeli and Jewish-Israeli adolescents who smoke.
Line 43: This reference is dated (2001). Will prefer more recent references for comparison.
Line 47: it will be helpful to provide information on the rate of smoking among the age group of interest in other countries in the middle east. The authors make a comparison with Europe. However, it may be challenging to make a real comparison given differences in lifestyle, culture etc.
Line 51: If it is already established that there is a higher rate of smoking among Arab-Israeli when compared with Jewish-Israeli, is there a reason to expect a difference among adolescents? If there is one , do highlight the reason why.
Discussion:
Overall, a nice discussion. Can be improved upon by expanding on the findings mentioned in the result section. For instance, It will be nice to provide a possible explanation for the decline in “every day smoking” in figure 2 c. The decline in 2006 especially for 10th graders occurs across ethnicity and gender. It will be helpful to be a little bit more detailed in discussing this finding. What factors may have been at play in 2006?
Author Response
Point 1: This paper examines trends in smoking among school -aged children in grades 6, 7 and 10 in Israel based on ethnicity and gender. Findings of this study can contribute to understanding characteristics of children who are likely to smoke cigarettes and tailor management efforts.
Authors make a good argument as to the need for this type of study and nicely show the trend in smoking rates by age and gender across a few years. This study highlights the differences among Arab-Israeli and Jewish-Israeli adolescents who smoke.
Line 43: This reference is dated (2001). Will prefer more recent references for comparison.
Response 1: Thank you, this reference was indeed dated we have updated it with a more current reference.
Point 2: Line 47: it will be helpful to provide information on the rate of smoking among the age group of interest in other countries in the middle east. The authors make a comparison with Europe. However, it may be challenging to make a real comparison given differences in lifestyle, culture etc.
Response 2: We believe that Israel is comparable to Europe on many levels, including smoking.
For example, Israel is included in the Health Behavior in School-aged Children study, which focuses only on European countries (and Israel).
Point 3: Line 51: If it is already established that there is a higher rate of smoking among Arab-Israeli when compared with Jewish-Israeli, is there a reason to expect a difference among adolescents? If there is one, do highlight the reason why.
Response 3: In the past decade there have been more intervention program and changes in policy on the subject of smoking among adolescents. Many of these programs have been implemented within the school setting.
This is a reason why we might expect deference in smoking rates among adolescents.
We added the following sentence to the introduction. In the past decade, "there has been an effort to implement such tobacco control programs in the school setting.”
Discussion:
Point 4: Overall, a nice discussion. Can be improved upon by expanding on the findings mentioned in the result section. For instance, It will be nice to provide a possible explanation for the decline in “every day smoking” in figure 2 c. The decline in 2006 especially for 10th graders occurs across ethnicity and gender. It will be helpful to be a little bit more detailed in discussing this finding. What factors may have been at play in 2006?
Thank you. We've included the following to the discussion: " Response 4: We found a decline every day smoking in 2006, which could possibly be due to the fact that in 2005, the Framework Convention on Tobacco Control was entered into force globally. This framework aims to reduce tobacco use, by implementing tax, advertising, and promotional restrictions on tobacco products.”
Please see the attachment.

Round 2
Reviewer 2 Report
No comments